# Choice of Ultrafilter Affects Recovery Rate of Bacteriophages

**DOI:** 10.3390/v15102051

**Published:** 2023-10-05

**Authors:** Frej Larsen, Simone Margaard Offersen, Viktoria Rose Li, Ling Deng, Dennis Sandris Nielsen, Torben Sølbeck Rasmussen

**Affiliations:** 1Department of Food Science, University of Copenhagen, 1958 Frederiksberg C, Denmark; lingdeng@food.ku.dk (L.D.); dn@food.ku.dk (D.S.N.); 2Section for Comparative Pediatrics and Nutrition, Department of Veterinary and Animal Sciences, Faculty of Health and Medical Sciences, University of Copenhagen, 1870 Frederiksberg C, Denmark; simo@sund.ku.dk (S.M.O.);

**Keywords:** virome, virus purification, VLP, bacteriophage, ultrafiltration

## Abstract

Studies into the viral fraction of complex microbial communities, like in the mammalian gut, have recently garnered much interest. Yet there is still no standardized protocol for extracting viruses from such samples, and the protocols that exist employ procedures that skew the viral community of the sample one way or another. The first step of the extraction pipeline often consists of the basic filtering of macromolecules and bacteria, yet even this affects the viruses in a strain-specific manner. In this study, we investigate a protocol for viral extraction based on ultrafiltration and how the choice of ultrafilter might influence the extracted viral community. Clinical samples (feces, vaginal swabs, and tracheal suction samples) were spiked with a mock community of known phages (T4, c2, Φ6, Φ29, Φx174, and Φ2972), filtered, and quantified using spot and plaque assays to estimate the loss in recovery. The enveloped Φ6 phage is especially severely affected by the choice of filter, but also tailed phages such as T4 and c2 have a reduced infectivity after ultrafiltration. We conclude that the pore size of ultrafilters may affect the recovery of phages in a strain- and sample-dependent manner, suggesting the need for greater thought when selecting filters for virus extraction.

## 1. Introduction

Historically, there has been a strong focus on the bacterial members of complex microbial ecosystems [1,2], but the extensive diversity and key functions of the viral component of such systems are becoming increasingly evident [3,4,5,6,7,8,9]. The viral fraction of these communities is, in general, dominated by the ubiquitous viral entity called bacteriophages (phages), while eukaryotic and archaeal viruses constitute the remaining fraction. Phages are viruses that, in a host-specific manner, infect bacteria to replicate and are therefore believed to play an important role in shaping the bacterial communities’ structure and stability in the respective environment [9,10,11,12,13]. Furthermore, recent studies have revealed that the human gastrointestinal tract harbors hundreds of different types of eukaryotic viruses with unknown functions [7,14,15] that remain to be studied and characterized. This emphasizes the importance of not just analyzing the bacterial, but also the viral component of microbial ecosystems. This can be carried out directly by investigating the viral fraction of metagenomic studies, finding viral contigs using bioinformatical tools such as VirSorter2 [16] or CheckV [17] as is achieved by Paez-Espino et al. [18]. However, most current studies choose to enrich the viral fraction of the sample prior to sequencing.

The first step of most viral enrichment pipelines is to separate the viral particles from bacteria and other contaminants. There is currently no accepted standard protocol, resulting in each phage research group having its own preferred methods [19,20,21,22]. Many pipelines include agitation to release the viral particles from the sample material and into a buffer, followed by centrifugation and filtration to remove larger particles and bacteria. The next step is then to concentrate the phages by decreasing the volume of the solvent and this is where most methods diverge; either using a tangential flow system [23,24], PEG-based precipitation, density gradient centrifugation, iron flocculation [25], ultrafiltration [21,26], or a combination of them.

There is still much discussion on the merits of each of the methods, and they all seem to bias the resulting viral community one way or another. Density gradient centrifugation loses phages outside the extracted density and some enveloped phages have been shown to lyse when exposed to cesium chloride [23,27,28,29]. Other viral taxa, such as herpesvirus and rotavirus, lose infectivity when purified using PEG-based precipitation [30,31], while ultrafiltration reduces the infectivity of tailed T4-like phages [22,32]. Additionally, the storage conditions and choice of buffer also seem to affect the infectivity of the phages, with SM-buffer being a better choice [19,33].

Ultrafiltration uses molecular-scale pores to concentrate particles, such as viruses, from a high-volume sample, while getting rid of smaller contaminants below the pore size that typically ranges between 10 and 300 kDa. It provides a relatively bias-free, cheap, and easily scalable method for producing analysis-ready viral communities, and the protocol developed by Deng et al. [22] provides a good basis for an analysis of the inherent biases introduced through ultrafiltration. In this study, we investigate the effects of a range of commercially available ultrafilter devices on the infectivity of an exogenous mock community of phages in both retentate and eluate in an array of different sample types.

## 2. Materials and Methods

### 2.1. Sample Collection

Human adult and infant samples were collected from healthy volunteers. Tracheal suction samples were collected at Gentofte Hospital and dissolved in 2 mL saline (0.9% (*w*/*v*) NaCl). Vaginal swabs were collected using a Copan ESwab [34] and dissolved in the accompanying medium. Human fecal samples were collected at home and stored at 4 ${}^{{}^{\circ}}$C until transport to the laboratory. Porcine feces were collected at a commercial farm (Bornholm, Denmark). All samples were transported to the laboratory within 24 h where they were diluted 1:1 in 2× SM buffer (400 mM NaCl, 20 mM MgSO_4_, 100 mM Tris-HCl, pH 7.5) containing 30% (*w*/*v*) glycerol and stored at −60 ${}^{{}^{\circ}}$C until extraction.

### 2.2. Phage Mock Community

A phage mock community was created from an in-house phage bank of virulent phages to represent a wide range of morphologically and taxonomically different types of phages (Table 1) commonly found in the tested sample types. The mock community had representatives from the three morphological groups previously known as *Siphoviridae*, *Podoviridae*, and *Myoviridae* [35], now in the families *Skunavirus*, *Brussowvirus*, *Tevenvirinae*, and *Salasvirus*, as well as a *Microviridae* species. Additionally, the *Pseudomonas* phage Φ6 was chosen to represent enveloped dsRNA phages. Although enveloped dsRNA phages are less commonly detected natively in the human gut microbiome [4,36], environmental and human RNA viruses are often present and could be important to the overall virome composition and activity as well as host health [37].

Phages were propagated individually by mixing 9.8 mL growth medium containing 10 mmol CaCl_2_ and 10 mmol MgCl_2_, 100 μL overnight host culture and 100 μL high concentration phage lysate in a tube and incubating at the temperatures indicated in Table 2. Bacteria requiring aeration were grown in wide conical flasks at 160 RPM. Growth media used were tryptic soy broth (TSB), lysogeny broth (LB, also known as Luria–Bertani broth), and M17 broth containing either 0.5% (*w*/*v*) lactose (LM17) or 0.5% (*w*/*v*) glucose (GM17). The following day, the lysates were centrifuged at 5000× *g* for 30 min and passed through a 0.45 μm filter to remove any remaining host bacteria, enumerated using spot assays (see Section 2.4.1), and stored at 4 ${}^{{}^{\circ}}$C until use.

The individual phages were mixed just before use to generate the phage mock community. This was done by transferring $5\times 10^{8}$ plaque-forming units (PFU) of each phage to a centrifuge tube and adding SM buffer (200 mM NaCl, 10 mM MgSO_4_, 50 mM Tris-HCl, pH 7.5) to the 50 mL mark, which resulted in an approximate concentration of $10^{7}$ PFU/mL for each phage.

### 2.3. Extraction of Viral-Like Particles

This procedure is identical to the one presented by Deng et al. [22]. Samples were thawed and 500 mg diluted feces, vaginal swab medium, or tracheal fluid was transferred to a 50 mL tube. They were spiked with 1 mL phage mock community and diluted with 29 mL SM buffer. The mix was homogenized using a stomacher 80 (Seward, Worthing, UK) for 2 min, centrifuged at 5000× *g* for 30 min at 4 ${}^{{}^{\circ}}$C, and the supernatant was then forced through a 0.45 μm Filtropur S polyethersulfone (PES) membrane filter (Sarstedt, Helsingborg, Sverige) mounted on a 20 mL syringe (Chirana, Stara Tura, Slovakia). The filtrated sample was then placed in the retention chamber of an ultrafiltration unit and centrifuged at 1500× *g* until less than 200 μL was left in the retention chamber. The eluate was saved for enumeration, while the phage-enriched retentate was diluted to 1 mL with SM buffer to equalize volumes across samples and incubated in the retention chamber at 4 ${}^{{}^{\circ}}$C overnight. See Figure 1 for an overview.

### 2.4. Enumeration of Infectious Phage Particles

Phages in the retentate of all filter/sample type combinations, as well as in the eluate of select samples, were enumerated for analysis using spot and plaque assays, respectively.

#### 2.4.1. Spot Assays

Soft agar (growth medium (see Table 2) containing 0.5% (*w*/*v*) agar) was melted in a microwave and brought to 50 ${}^{{}^{\circ}}$C in a water bath. Square petri dishes (81 cm^2^, Simport, Saint-Mathieu-de-Beloeil, QC, Canada) with 40 mL hard agar (growth medium containing 1.2% (*w*/*v*) agar) were brought to room temperature. 10 mmol CaCl_2_ and 10 mmol MgCl_2_ (concentrations in final solution) was added to 5 mL soft agar which was then mixed with 500 μL overnight host culture and quickly poured on top of the hard agar plates. This was repeated for each of the host strains. Retentate from the virus-like particle (VLP) extraction was then diluted from 10^−1^ to 10^−6^ times in SM buffer and 5 μL of each fraction was spotted on the bacterial lawn of the square dishes using a multichannel pipette. The spots were dried out on the table with the lid ajar for an hour and were then incubated overnight at the temperatures described in Table 2. The following day, a spot with 5 to 20 plaques was enumerated in order to estimate the concentration of phages in the original lysates.

#### 2.4.2. Plaque Assays

Like for the spot assays, soft agar was liquefied and brought to 50 ${}^{{}^{\circ}}$C in a water bath. Round petri dishes (61 cm^2^) containing 25 mL hard agar were brought to room temperature. Amounts of 100 μL undiluted eluate and 300 μL overnight host culture were gently mixed and incubated for 10 min. An amount of 10 mmol CaCl_2_ and 10 mmol MgCl_2_ along with 3 mL soft agar was added to the mixture, which was quickly mixed and poured evenly onto the petri dishes. Once the soft agar had hardened, the dishes were inverted and incubated overnight, similar to the spot assays. For plaque assays, a maximum of 200 plaques was counted per plate, giving a PFU/mL bounded by 10 and 2000 PFU/mL.

## 3. Results

### 3.1. Ultrafiltration

The filters tested in this study were selected to have properties similar to the CentriPrep device used by Deng et al. [22], which is currently out of production, such as membrane material and pore size (Table 3). Their molecular weight cut-off (MWCO) was 30 kilodalton (kDa), except for the Centrisart device, which had an MWCO of 100 kDa. A pore size of 100 kDa roughly translates to a radius of 3 nm for spherical proteins [38]. As this is below the lowest found size of phages [39], smaller molecules such as DNA fragtments and metabolites can pass through, while the phages stay in the retention chamber of the filter device. The membranes of the Vivaspin, Pierce Protein Concentrator, and the Centrisart filter devices are made of polyethersulfone (PES), while the membrane of the Amicon Ultra-15 device, as well as the original CentriPrep device, have membranes made of regenerated cellulose. PES is, however, very close to cellulose in properties, as they are both hydrophilic membranes with the low binding of both proteins and DNA, providing a high flow-through with little clogging [40,41].

Three of the filters have a similar volume capacity to that of CentriPrep and are capable of processing around 15–20 mL at a time, whereas the Centrisart has a capacity of 2.5 mL. The CentriPrep and Centrisart filters utilize reverse flow; the sample is placed in the bottom chamber of the device and centrifugation forces the liquid up through the filter membrane and into the elution chamber above (see Figure 2). The reverse design keeps large molecules remaining in the solution from clogging the filters as they are instead forced to the bottom of the retention chamber. However, when compared to the standard flow devices, they have an increased hands-on time due to the lower volume capacity and more elaborate opening mechanisms.

### 3.2. Rate of Recovery in Retentate

The phage mock community was mixed with the sample matrices as a quantifiable proxy of the real viral community present. The sample, along with its spiked phage community, was passed through the VLP extraction pipeline (Section 2.3), and the number of plaque-forming units was determined using spot assays and compared to the original concentration of the phage mock community. This ratio between the input phage and the resulting infective units was used to quantify the rate of recovery (ROR) of infective units across sample matrices (human tracheal suction, vaginal swab, and adult, infant, and pig feces) and filters (VivaSpin, Amicon, CentriPrep, Pierce, and CentrisArt) for the six phages in the phage mock community (T4, c2, Φ6, Φ29, Φx174, and Φ2972).

Most combinations of sample matrix/filter/phage ended up with an ROR at approximately 10^0^ (100%), suggesting that most phages are fully recovered after ultrafiltration. The accuracy of spot assays is slightly reduced compared to plaque assays, as quantification is based on a smaller volume (10 µL versus 100 µL) as well as an increased chance of overlapping infections. As such, all results above 10^−1^ (10% ROR) are considered fully recovered. This is illustrated in the results (Figure 3 and Figure 4) by a green span.

Extracting phages from samples with phage mock, but no added sample matrix, resulted, for most phage/ultrafilter combinations, in an ROR between 10^0^ and 10^1^ (10–100% of input phage). The ROR is particularly low for the tailed *Skunavirus* phages c2 (Figure 3f) and Φ2972 (Figure 3b) when extracted using the Amicon ultrafilter. Interestingly, samples with an added sample matrix did not have a decreased ROR, even with the Amicon device.

The drop in infectivity of the tailed phages was also remarked upon by Deng et al. [22]. They, however, only observed a reduced ROR for the T4 phage and not the c2 *Skunavirus* phage, despite using the same strain as in this current study. They explain the reduced ROR of the T4 phage by its rigid tail structure being vulnerable to centrifugation, however, this is not what we see in this study, as the ROR of T4 in most sample types is similar to that of the globular *Microviridae* Φx174 and the tailed, but smaller, *Salasvirus* phage Φ29. We do, however, see a significant drop in T4 phage in the infant fecal sample type when we—similar to Deng et al.—use the CentriPrep filter, suggesting that it is not the centrifugation step causing the reduction in ROR, but rather an unknown factor in the sample material of the filtration unit that inhibits the infectivity of the phage.

For the fecal matrices, the ROR is mostly between 10^1^ and 10^0^. The short-tailed *Salasvirus* Φ29 and the spherical *Microviridae* are close to 100% recovery, while the tailed phages are slightly lower (mean recovery: T4: 67%, C2: 75%, Φ2972: 41%) and the ROR of the enveloped *Cystoviridae* phage Φ6 is severely impacted by both filter type and sample matrix. In tracheal, vaginal swab (Figure 3e), and pig fecal (Figure 4e) sample matrices, the recovery of Φ6 is between 10^−1^ and 10^0^ (10–100%). However, in human feces (both adult and infant), the ROR is reduced below 10^−4^ in the Amicon filter, translating to a recovery of less than 0.01% of the input phage. In sharp contrast, the Φ6 phage has a ROR of 10^−1^ when using the Centrisart filter, suggesting that the severe reduction in recovery in the other filters is due to the combination effect of both sample matrix and filter type.

### 3.3. Phage Presence in Eluate

To explore the loss in phage infectivity, we measured the concentration of infective phages in the eluate after ultrafiltration to see whether some phage particles were able to pass through the filters. The phage-free eluates might also be of interest as a negative control in fecal viral transplantation or metabolite transplantation experiments, in which case the presence or absence of viruses in the eluate is vital.

Plaquing 100 μL of eluate from the pig fecal samples on the phage hosts revealed that none of the ultrafilters produced 100% phage-free eluates. CentriPrep leaked the most phages into the eluate, reaching the counting threshold of 2000 PFU/mL (Figure 5). Centrisart had Φx174 and c2 present in all replicates (Figure 5). The reverse flow design can possibly explain the increased phage concentration of these two filters as the eluate chamber is nested within the retention chamber and thus likely increasing the risk of contamination during sampling. There was no discernible association between phage concentrations in the eluate (2000 PFU/mL and above) and a lowered concentration in the retentates from the same filters (Figure 4). Eluates from Vivaspin and Pierce had the lowest contamination levels, with only a few phage particles present in the range of 10–100 PFU/mL—less than 0.001% of the input phage titer—and thus these might be the most optimal filters for generating phage-free eluates.

## 4. Discussion

In this study, we investigated the effects of a range of sample types and ultrafilters on the ROR of a phage mock community representing viral families most commonly found in human samples [42]. We found that the ROR was highly dependent on both sample and filter type, and that even small changes, such as whether the mock community was mixed with adult or infant fecal matter, affected the ROR for some phages. The enveloped Φ6 phage was especially heavily affected, with highly reduced infectivity after filtration in a human fecal matrix. This could be due to enveloped phages being more sensitive to the laboratory environment [27], or possibly due to phage-inactivating contaminants present in the fecal matter [43,44].

The difference in the ROR of the Φ6 phage between filters with different pore sizes (Figure 4e) corroborates this, as the increased MWCO of the Centrisart filter (Table 3) allows larger particles to pass through the filter and away from the retentate during ultrafiltration. Such contaminants could, for instance, be lipases or lysozymes that dissolve the membrane surrounding the phage particle, or cellular debris, such as membrane proteins or bacterial pili, which are capable of binding or aggregating the Φ6 phages, rendering them unable to infect their hosts in the spot assay [44,45,46,47]. This source of bias would be even more pronounced for endogenous phages than for phages from a mock community, as their natural hosts would likely be present in the sample, increasing the risk of residual bacterial debris in the retentate capable of disrupting the phages.

A third explanation could be that phages were able to pass through the ultrafilters, potentially aided by some agent in the sample matrix. However, the plaquing efficiency of the eluate did not correspond to the loss in ROR, which would be expected if some filter/sample combinations were more prone to pass phages through the membrane than others. Instead, it seemed that the reverse filters specifically had a higher concentration of phages in the eluate than those with a standard direction of flow (Figure 5), most likely caused by contamination from the retention chamber due to their proximity and the open design of the retention chamber.

With the increased interest in the gut virome, several studies have investigated the community bias created during the extraction and DNA processing process:; Conceição-Neto et al. produced a thorough protocol for VLP extraction, finding that the pore size of ultrafilters has a significant effect on both bacterial contamination and the abundance of specific viral species in the eluate, as well as how chloroform treatment, bead homogenization, and centrifugation speed affect the viral community [21]. Pérez-Cataluña et al. investigated the effect of different combinations of nucleases on irrigation-water-derived viral samples [48] and Shkoporov et al. developed a pipeline that produces virome data with very low levels of bacterial contamination and explored the effects of freeze–thaw cycles as well as operator reproducibility [19]. The studies mentioned all quantify their results using either qPCR or sequencing, which are valid methods for virome studies.

Although the genetic material may be present, the phages may have lost their infectivity through the breaking of the phage tail or membrane rupture [22,23], which could be of great importance for isolating novel culturable phages or for concentrating phages for therapeutic purposes [49]. Recent examples of this include studies on fecal virome transplantations (FVT) where filtered stool from healthy donors is used to treat diseases caused by the dysbiosis of the gut, such as necrotizing enterocolitic [50], *Clostridioides difficile* infection [51,52], type-2-diabetes and obesity [8], or to manipulate the native gut community [52]. Here, the continued infectivity of the viral community after filtration and concentration is essential, and, thus, studies investigating how the infective community is skewed in the lab will be important for these promising treatment options.

This study is a refinement of a previous validated protocol [22] that provides an easy, cost-efficient, and scalable method for extracting VLPs from environmental samples for downstream applications such as plaque assays or sequencing. The immense diversity of viruses in term of sizes, morphology, sensitivity, and binding capabilities makes bias-free protocols nearly impossible. Our investigation focused on how the choice of ultrafilter would affect the rate of recovery (ROR) of phages, while other design choices, such as the choice of buffer [33], pore size of the membrane filter [21], or temperature [44,53] also affects the ROR in a species-dependent manner. In this current protocol, a pore size of 0.45 μm was used for sterile filtration to ensure that the majority of larger viruses were included in the VLP-enriched fraction for downstream analysis. However, as a consequence, small bacterial species such as Polynucleobacter and some Actinobacteria [54,55] may also pass through the filter. Choosing a smaller pore size (such as the 0.22 μm filter) can minimize this contamination, but will cause the exclusion of viruses above 220 nm in diameter, such as giant viruses and larger phages [56,57].

Even with the increased effort in creating extraction protocols for culturing and virome analysis, we are still far away from a bias-free method. Yet, these studies give us an understanding of the mechanics of phage loss, of how the community is affected, and thus how protocols can be adapted to better suit the specific experimental purposes.

## Figures and Tables

**Figure 1 viruses-15-02051-f001:**
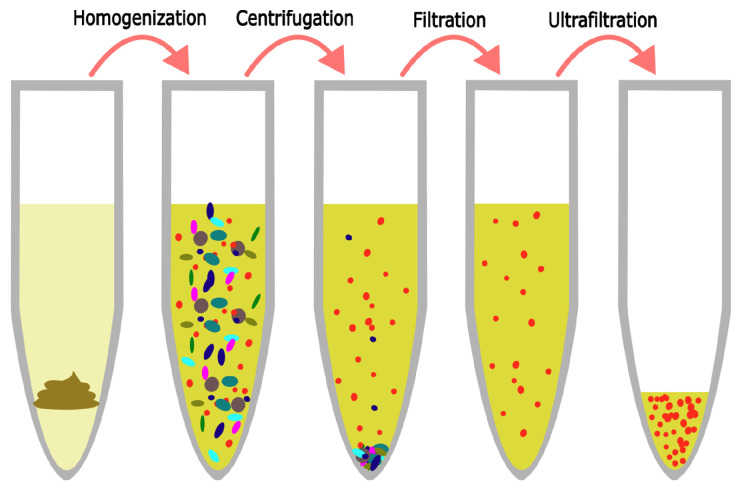
The VLP extraction protocol consists of 4 primary steps. The sample is first dissolved in SM buffer to release viruses into the buffer. Centrifugation pellets bacteria and macromolecules. Filtration removes remaining bacterial contaminants. Ultrafiltration concentrates phages in a small volume.

**Figure 2 viruses-15-02051-f002:**
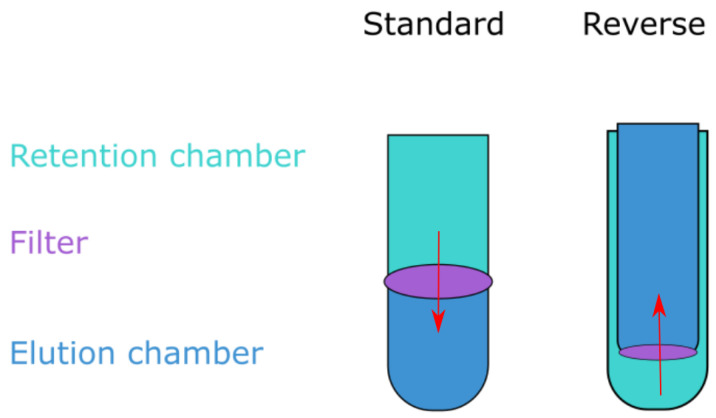
Ultrafiltration unit types. Ultrafiltration devices come in two formats: either the retention chamber is on the top and the sample is pushed down through the filter into the bottom elution chamber (standard), or the retention chamber is on the bottom and the sample is forced up through the filter membrane by centrifugation (reverse). Any sample left in the retention chamber is the retentate, while liquid passing through is coined the eluate.

**Figure 3 viruses-15-02051-f003:**
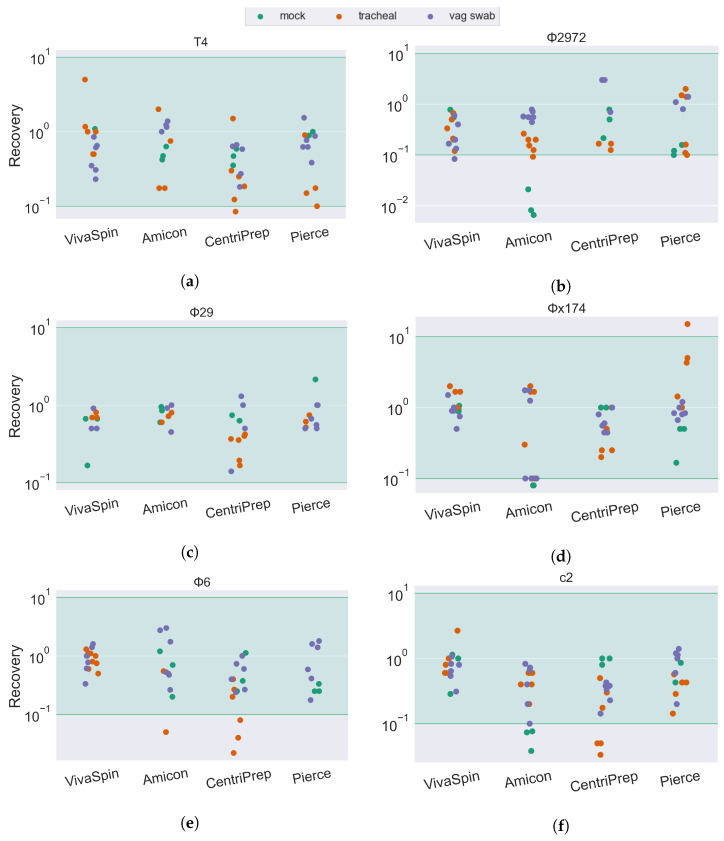
Bacteriophage recovery rate of 6 different phages forming a mock community in SM buffer, in tracheal suction, and in a vaginal swab sample. Recovery rates are calculated as the concentration of infective phages in the retentate divided by the concentration of infective phages in the initial phage mock community. The green area is considered full recovery. Subfigures correspond to individual phage strains: (**a**) T4, (**b**) Φ2972, (**c**) Φ29, (**d**) Φx174, (**e**) Φ6, and (**f**) c2.

**Figure 4 viruses-15-02051-f004:**
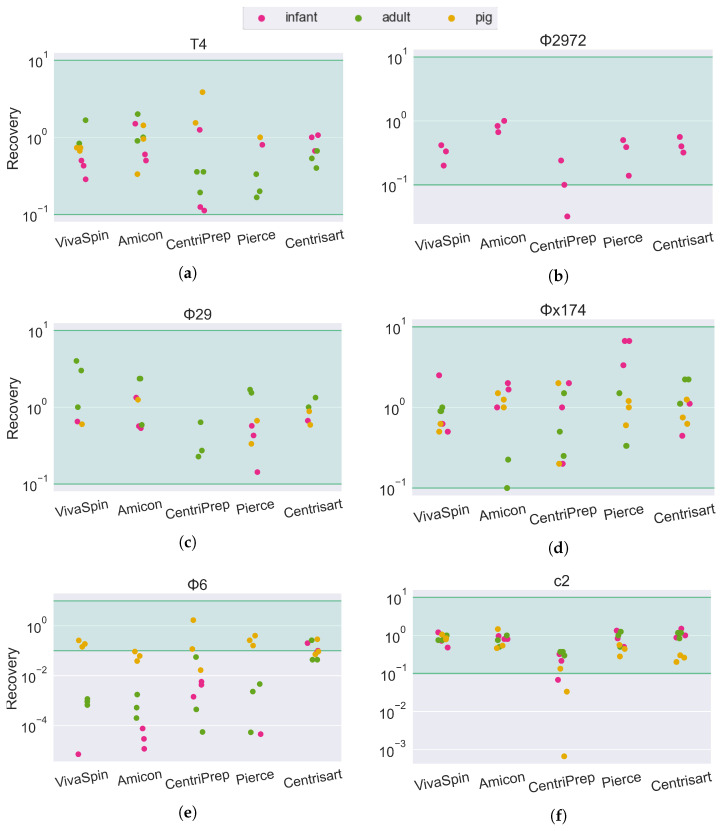
Phage recovery rate of 6 different phages forming a mock community in feces from human infants, human adults and colon content from 10-day-old pigs. Recovery rates are calculated as the concentration of infective phages in the retentate divided by the concentration of infective phages in the phage mock community. The green area is considered full recovery. Subfigures correspond to individual phage strains: (**a**) T4, (**b**) Φ2972, (**c**) Φ29, (**d**) Φx174, (**e**) Φ6, and (**f**) c2.

**Figure 5 viruses-15-02051-f005:**
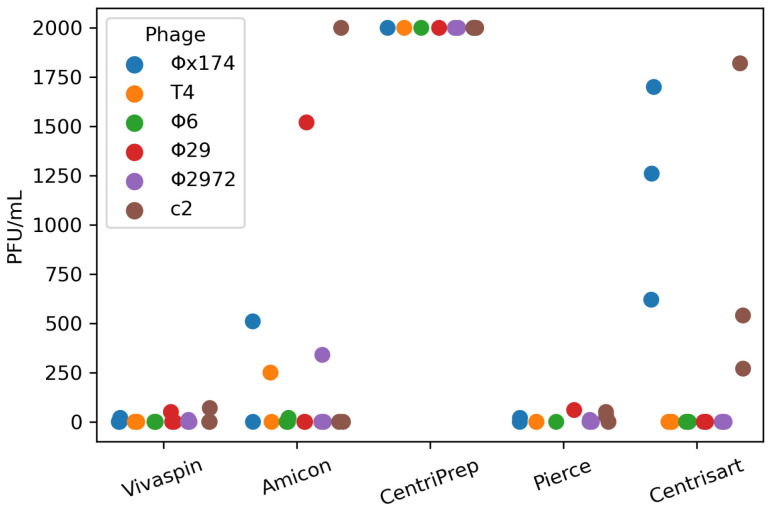
Phage concentration in ultrafilter eluate from a pig fecal matrix measured in PFU/mL for 5 different ultrafilters. The upper detection limit is 2000 PFU/mL. The lower limit is 10 PFU/mL.

**Table 1 viruses-15-02051-t001:** Properties of phages used in mock community. Dimensions are given in diameter (ø) for spherical phages, and head diameter/tail length for tailed phages. All measurements are in nanometers (nm).

Name	Family	Genome	Dimensions (nm)	Morphology
Φx174	*Microviridae*	ssDNA	ø50	Icosahedral
Φ6	*Cystoviridae*	dsRNA	ø70	Icosahedral, Enveloped
c2	*Skunavirus*	dsDNA	40/150	Long, non-contractile tail
Φ2972	*Brussowvirus*	dsDNA	50/250	Long, non-contractile tail
T4	*Tevenvirinae*	dsDNA	90/200	Long, contractile tail
Φ29	*Salasvirus*	dsDNA	41/30	Short tail

**Table 2 viruses-15-02051-t002:** Phage host growth parameters. Growth medias are lysogeny broth (LB), tryptic soy broth (TSB), and M17 broth containing either 0.5% (*w*/*v*) lactose (LM17), or 0.5% (*w*/*v*) glucose (GM17). Host/phage pairs requiring aeration were incubated in a shaking incubator moving at ∼160 RPM.

Phage	Host Species	Host Strain	Growth Media	Temperature	Aeration
Φx174	*Escherichia coli*	ATTC13706	LB	37	Yes
Φ6	*Pseudomonas* sp.	DSM21482	TSB	25	Yes
c2	*Lactococcus lactis*	MG1363	GM17	30	No
Φ2972	*Streptococcus thermophilus*	DGCC7710	LM17	42	No
T4	*Escherichia coli*	DSM613	LB	37	Yes
Φ29	*Bacillus subtilis*	DSM5547	TSB	37	Yes

**Table 3 viruses-15-02051-t003:** Concentrators and their properties and distributors. The molecular weight cut-off (MWCO) describes the pore size of the filters; molecules above the given value in kilodalton (kDa) cannot pass through. The membrane column describes the filter membrane material, which is either regenerated cellulose (RC) or polyethersulfone (PES). The volume metric describes the size of the retention chamber, and thus the amount of sample that can be added at a time.

Name	MWCO (kDa)	Volume (mL)	Membrane	Flow
Vivaspin	30	20	PES	Standard
Pierce Protein Concentrator	30	20	PES	Standard
Amicon Ultra-15	30	15	RC	Standard
CentriPrep	30	15	RC	Reverse
Centrisart	100	2.5	PES	Reverse

## Data Availability

Plaque counts are available at https://doi.org/10.17605/OSF.IO/N84D7, accessed on 4 October 2023.

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
