# Peer review of "Choice of Ultrafilter Affects Recovery Rate of Bacteriophages"

_viruses, 2023, doi:10.3390/v15102051_

Round 1
Reviewer 1 Report
The article "Choice of Ultrafilter Affects Recovery Rate of Bacteriophages" picks a critical topic to study the microbiome of mammals including humans. With this study, the authors have narrowed to ultrafiltration as the best choice to concentrate viruses. Yet, bias is still noticeable, but is a great step toward standardization. In that case, the authors should include in the discussion section the bias the spot assay method might also carry.
Also, I suggest that in the material and methods section, specifically in subsection 2.3, to better describe the extraction of viral-like particles, a diagram will guide the readers step by step.
Other than that, I am satisfied with the narrative and the experimental design.
Author Response
Please find attached response to the reviewers.

Reviewer 2 Report
In this study, Larsen et all compare the effect of different types of ultrafilter units to process a mock viral community mixed with different biological samples. Unfortunately, despite the study is well conducted, there is no a major conclusion or recommendation in terms of a better protocol to follow. But, we should not blame the authors, it is simply the result of a complex issue since phages are extremely diverse and matrices of biological samples as well. Altogether, this makes very difficult to find like a universal protocol or a "perfect filter" for all applications. Authors have tried to assessed that, and have found like many others in the past (see Sullivan´s group or others leading the field in environmental virology), that there is no a perfect method.
Major concern:
In this study, and the same in the one performed by Deng et al cited several times in the manuscript, authors perform basically a pre-filtration of sample by 0.45 um PES filter, and then ultra concentrate that fraction by using different ultrafilter units (Centriprep, etc..). Nothing new by the way. There are a lot of publications using similar protocol. The point here, is that they try to compare the performance of different filters, which is a good point. After concentration, then, they work with this concentrated samples, called by authors like "purified virus fraction" or "extraction of viral-like particles" or similar sentences. THIS IS NOT CORRECT, and indeed when you pre-filter a seawater sample, or fecal sample, or a freshwater sample from a lake by 0.45 um , you get a lot of small bacteria in your elute. This is absolutely normal!. Thus, if you concentrate that fraction containing small bacteria, such as Pelagibacter in seawater, Polynucleobacter or Actinobacteria acI from freshwater, ACTUALLY you do NOT have a pure viral fraction but it is already naturally contaminated with bacteria. Authors do not discuss that in the text, which is a major issue and concern. It seems, that as is presented, that the purified fraction is "perfectly" pure only containing phages, but this is absolutely NOT CORRECT. If authors want to test that, I encourage you to do a SYBR Gold stain of the purified concentrated sample, and they will see bacteria as well. 100% positive. If you sequence that fraction, you will find sequences from bacterial genomes as well. Unless paper does not address this question and discuss this bias, manuscript should not be accepted.
Minor concern
Authors have to acknowledge in the introduction that not all researchers perform a previous separation and purification of viruses. There are great examples of surveys in which directly from meta genomics, authors use bioinformatics to find and detect viral contains. This methodology is extremely powerful and I recommend to care and explain as well this other way. See for instance "Uncovering Earth's virome" paper...or papers by Rodriguez-Valera´s group or many others. These are just some examples
Introduction ln 19...reference needed.
Ln24 modify to "hundreds of different types of eukaryotic viruses"...much better
I miss a definition or explanation of what is "ultrafiltration" and what does it mean here...please explain it right away in the introduction.
Author Response

(The authors gave the same response as above.)
